# Transforming Growth Factor-β Signaling in Fibrotic Diseases and Cancer-Associated Fibroblasts

**DOI:** 10.3390/biom10121666

**Published:** 2020-12-12

**Authors:** Xueke Shi, Christian D. Young, Hongmei Zhou, Xiao-Jing Wang

**Affiliations:** 1State Key Laboratory of Oral Diseases, National Clinical Research Center for Oral Diseases, Department of Oral Medicine, West China Hospital of Stomatology, Sichuan University, Chengdu 610041, Sichuan, China; xueke.shi@cuanschutz.edu; 2Department of Pathology, University of Colorado Anschutz Medical Campus, Aurora, CO 80045, USA; christian.young@cuanschutz.edu; 3Veterans Affairs Medical Center, VA Eastern Colorado Health Care System, Aurora, CO 80045, USA

**Keywords:** TGF-β signaling, fibroblasts, fibrotic diseases, cancer, cancer-associated fibroblast, anti-fibrosis/cancer therapy

## Abstract

Transforming growth factor-β (TGF-β) signaling is essential in embryo development and maintaining normal homeostasis. Extensive evidence shows that TGF-β activation acts on several cell types, including epithelial cells, fibroblasts, and immune cells, to form a pro-fibrotic environment, ultimately leading to fibrotic diseases. TGF-β is stored in the matrix in a latent form; once activated, it promotes a fibroblast to myofibroblast transition and regulates extracellular matrix (ECM) formation and remodeling in fibrosis. TGF-β signaling can also promote cancer progression through its effects on the tumor microenvironment. In cancer, TGF-β contributes to the generation of cancer-associated fibroblasts (CAFs) that have different molecular and cellular properties from activated or fibrotic fibroblasts. CAFs promote tumor progression and chronic tumor fibrosis via TGF-β signaling. Fibrosis and CAF-mediated cancer progression share several common traits and are closely related. In this review, we consider how TGF-β promotes fibrosis and CAF-mediated cancer progression. We also discuss recent evidence suggesting TGF-β inhibition as a defense against fibrotic disorders or CAF-mediated cancer progression to highlight the potential implications of TGF-β-targeted therapies for fibrosis and cancer.

## 1. Introduction

TGF-β is known to participate in various cellular processes, including differentiation, proliferation, migration, extracellular matrix (ECM) remodeling, and apoptosis, all of which influence embryogenesis, wound healing, fibrosis, inflammation, and tumor progression [1]. TGF-β ligands consisting of TGF-β1, TGF-β2, and TGF-β3 are secreted by numerous cell types, including epithelial cells, fibroblasts, and immune cells [2,3], and stored in the tumor microenvironment (TME, the environment surrounding the tumor) in an inactive form [4]; activated TGF-β ligands exert their role through an autocrine or paracrine manner [5]. Activated TGF-β dimers interact with TGF-β type II receptors (TβRII) that recruit TGF-β type I receptors (TβRI), inducing transphosphorylation of the receptor complex. Activated TβRI phosphorylates small mothers against decapentaplegic homolog (SMAD)2 and SMAD3 (R-SMADs); phosphorylated SMAD2 and SMAD3 (p-SMAD2 and p-SMAD3) oligomerize with SMAD4 and translocate to the nucleus to regulate expression of TGF-β target genes (Figure 1). This R-SMADs dependent pathway is referred to as canonical TGF-β signaling. SMAD7 is a negative feedback inhibitor of TGF-β/SMAD canonical signaling that physically compete with canonical SMAD proteins, as well as recruit E3 ubiquitin ligases to induce proteaosomal degradation of TGF-β signaling components to dampen TGF-β/SMAD signaling [6,7]. TGF-β also induces non-SMAD signaling (non-canonical pathways), including Rho GTPases (Rho), mitogen-activated protein kinases (MAPK), phosphoinositide-3-kinase (PI3K), and p53 [8,9]. These non-SMAD signaling pathways are involved in TGF-β-mediated biological responses, and they can also regulate the canonical SMAD pathway [10,11,12,13,14]. (Figure 1)

Fibroblasts are mesenchymal cells, a major cell type producing ECM proteins that are generally quiescent in non-pathological adult tissues. However, they can become activated myofibroblasts via TGF-β signaling after an injury to produce a different set of ECM molecules during wound healing and tissue inflammation, thus playing a critical role in wound healing and resolution [15]. Activated TGF-β signaling not only directly promotes fibroblast activation and proliferation, but also acts on epithelial cells to induce fibrotic factors [16,17], ultimately leading to fibrotic diseases through increased ECM production [18], as discussed in greater detail below.

Cancers can be thought of as “wounds that never heal”, i.e., a disease state with chronic inflammation [19,20]. As we and others have noted, TGF-β signaling plays a paradoxical role in cancer; it has anti-tumor and pro-tumor effects in the early and late stages of cancer progression, respectively [21,22]. Activated fibroblasts in cancer, termed “cancer-associated fibroblasts” (CAFs), are among the most abundant cell types in the cancer stroma. TGF-β signaling induces diverse changes in CAFs (and other cell types), supporting tumor growth and survival [23]. Compared to normal fibroblasts, CAFs secrete more growth factors and cytokines, including TGF-β. CAFs can crosstalk with cancer cells and other stromal cells via TGF-β ligands, thus advancing tumor development through the establishment of a favorable TME for cancer growth [24].

In this review, we focus on TGF-β signaling in fibrotic diseases and CAFs, and the reciprocal interaction of CAFs with cancer cells via TGF-β signaling. We also elaborate on potential therapeutics targeting TGF-β signaling to more effectively treat patients with fibrotic disease and cancer.

## 2. TGF-β Signaling and Fibrotic Diseases

### 2.1. Introduction to Fibrotic Diseases

Every organ in the body can be affected by physiologic and pathologic fibrotic reactions. These reactions are self-limited under homeostatic conditions and resolve damage in injured tissues, a process critical to health and life. However, under chronic and persistent pathologic circumstances, an uncontrolled fibrotic process results in excessive accumulation of ECM, including collagen, fibronectin, hyaluronic acid, and proteoglycans that leads to fibrosis and ultimately organ failure in fibrotic diseases [25,26,27]. These fibrotic diseases may occur in multi-system diseases, such as systemic sclerosis (SSc) [28], nephrogenic systemic fibrosis (NSF) [29], and specific organs like idiopathic pulmonary fibrosis (IPF), renal fibrosis, hepatic cirrhosis, cardiac fibrosis, oral submucous fibrosis, and glaucoma [30,31].

Although mechanisms of human fibrotic diseases are diverse, all fibrotic diseases have molecular changes that contribute to the accumulation of excessive collagen and other ECM components, replacing normal tissues with nonfunctional fibrotic tissue [32,33,34,35]. For example, hepatic fibrosis is the consequence of a persistent wound-healing process. After an acute liver injury, parenchymal cells regenerate and replace the necrotic or apoptotic cells. This process is associated with a limited deposition of ECM. If the hepatic injury persists chronically, liver regeneration fails, and hepatocytes are replaced with abundant ECM, including fibrillar collagen, impeding normal liver function [32]. At the cellular level, increased synthesis of ECM is determined by activation of ECM-producing cells, namely, myofibroblasts or fibrosis-associated fibroblasts (FAFs, myofibroblasts associated with fibrosis) [36]. FAFs express alpha-smooth muscle actin (α-SMA), produce vast quantities of ECM proteins, and are the key pathogenic cell type in all fibrotic diseases [30,37,38,39,40]. Other cells like epithelial cells and immune cells also participate in the fibrosis via the production of fibrotic factors to exert paracrine signals [16,41], suggesting that different cell types participate in the fibrosis cascade (Figure 2).

Several signaling pathways are involved in fibrosis [18,42,43,44], but TGF-β signaling is the master regulator of fibrosis [18]. TGF-β ligands or downstream stimulating effectors are overexpressed in lesions of human fibrotic diseases, such as SSc, cardiac fibrosis, and NSF [45,46,47]. Animal models demonstrated that TGF-β activation induces fibroblasts to produce excessive ECM, causing fibrosis [48]. Additionally, TGF-β signaling in non-fibroblast cells can also induce fibrosis via the production of fibrotic factors, such as endothelin 1 (ET-1), connective tissue growth factor (CTGF), interleukin (IL)-13, platelet-derived growth factor (PDGF), fibroblast growth factor (FGF-2), and insulin-like growth factor (IGF)-1/2 [13,14,16,49,50]. In the following section, TGF-β signaling-induced cellular responses and molecular changes in fibroblasts and non-fibroblasts will be briefly reviewed to clarify how TGF-β activation results in fibrosis (Figure 2).

### 2.2. TGF-β Signaling in Fibroblasts and Fibrosis

#### 2.2.1. TGF-β Signaling and Myofibroblast Generation

Myofibroblasts are key pathogenic cells in human fibrotic diseases, and TGF-β is overexpressed in human fibrotic diseases, such as cardiac fibrosis and NSF [51,52]. After an injury or during chronic inflammation, spindle-shaped quiescent fibroblasts can be activated and differentiated to stellate-shaped myofibroblasts that have high contractility [15,36]. It is reported that TGF-β signaling contributes to fibroblast-myofibroblast transdifferentiation through activation of both SMAD and non-SMAD signaling pathways [50]; the following sections will elucidate how these signal transmissions influence the transdifferentiation of fibroblasts into myofibroblasts.

The canonical TGF-β pathway is involved in myofibroblast transdifferentiation, as shown by increased levels of p-SMAD2 and p-SMAD3 during TGF-β1 induced myofibroblast generation from human skin fibroblasts [53]. Khalil et al. found that fibroblast-specific deletion of *Tgfbr1, Tgfbr2*, or *Smad3*, but not *Smad2*, significantly reduced α-SMA expression in mouse skin fibroblasts following stimulation of TGF-β1 in vitro and markedly reduced fibrosis in a cardiac mouse model [48]. This study indicates that TGF-βRI, TGF-βRII, and SMAD3, but not SMAD2 are critical mediators of myofibroblast differentiation and fibrosis (Figure 1).

Active R-SMADs can also interact with other transcription factors (co-activators/co-repressors) in the nucleus to regulate myofibroblast generation (Figure 1). For instance, activated SMAD3 assembles with p53, and the p53-SMAD3 complex is required for transcription of TGF-β induced fibrotic genes, such as plasminogen activator inhibitor-1 (PAI-1) and CTGF in renal fibrosis [9]. p53 inhibitor attenuated TGF-β1 induced morphological transformation in renal fibroblasts and reduced expression of α-SMA. Additionally, PAI, CTGF, and fibronectin were all reduced by p53 inhibitor treatment. These observations implicate p53 as a co-activator with p-SMAD3 to promote TGF-β1-induced myofibroblast generation and fibrotic gene expression, leading to fibrosis [9,54] (Figure 1). Another example of non-canonical transcription factors cooperating with SMADs are Hippo signaling effectors Yes-associated protein (YAP)/transcriptional co-activator with PDZ-binding motif (TAZ) cooperating with p-SMAD2/3 in driving the renal fibrosis [55]. YAP/TAZ are mechanosensors that can bind to R-SMADs [55]. YAP/TAZ accumulate in cytoplasm or nuclei in cells grown on soft or stiff substrates, respectively, indicating stiff substrates control these transcription factors’ subcellular localization, and therefore, activity [56]. Rat kidney fibroblasts cultured on stiff ECM have stronger TGF-β–induced pro-fibrotic responses with increased p-SMAD2/3 nuclear translocation in a process that is mediated by YAP/TAZ [55]. Downregulation of Yap/Taz in fibroblasts prevents the nuclear translocation of p-SMAD2/3 complexes and blocks the expression α-SMA in fibroblasts. CTGF is also decreased with YAP/TAZ downregulation [55], demonstrating that the transcription factors YAP/TAZ in fibroblasts also cooperates with TGF-β canonical signaling to promote fibrosis. These findings reveal that non-canonical TGF-β signaling “crosstalk” with the canonical SMAD pathway during the process of myofibroblast generation and fibrosis through the formation of co-transcription factor units between active R-SMADs and components in non-canonical TGF-β signaling.

Non-canonical signaling can also drive myofibroblast transdifferentiation without the influence or participation of activated R-SMADs. Phosphorylation of MAP-kinase ERK1/2 was consistently elevated following TGF-β stimulation of human skin fibroblasts, and ERK1/2 activation promoted differentiation of fibroblasts via the upregulation of transcription factor FRA2, a downstream mediator of TGF-β with a pro-fibrotic effect [57,58] (Figure 1) [53]. Blocking ERK1/2 activation with MEK1/2 inhibitor can attenuate TGF-β-mediated activation of myofibroblasts with no effect seen on p-SMAD2 or p-SMAD3 [53]. Another example is when Rho signaling on actin formation/re-organization is disrupted, myofibroblast generation is attenuated, as is collagen synthesis in human lung fibroblasts [59]. A mouse in vitro model using Swiss3T3 fibroblasts further demonstrated that activated TβRI stimulated Rho GTPases via Rho-Rho Kinase 1-LIM-kinase 2 phosphorylation that inactivated the actin-depolymerizing factor cofilin, leading to actin re-organization (Figure 1) [60]. SMAD2/3 mutation in Swiss3T3 fibroblasts did not affect actin remodeling or Rho activation [60], indicating that non-canonical Rho signaling components, but not R-SMADs of canonical TGF-β signaling are responsible for actin formation and re-organization. In this study, SMAD7 ectopic overexpression inhibited TGF-β1 induced Rho activation and actin re-organization in Swiss3T3 fibroblasts (Figure 1) [60], demonstrating that SMAD7 exerts an inhibitory role for this critical step of actin re-organization during fibroblast transdifferentiation. These examples shed light on how non-canonical TGF-β signaling can induce myofibroblast activity without canonical R-SMAD signaling.

#### 2.2.2. Role of TGF-β in Proliferation and Apoptosis of Fibroblasts

Fibroblast foci are widely scattered and can be found in human fibrotic areas, indicating that fibroblast proliferation is strengthened in fibrotic diseases [61]. Meran et al. revealed that TGF-β1 enhanced human dermal fibroblast proliferation via increased SMAD2 and SMAD3 phosphorylation. SMAD3 downregulation results in abrogated proliferation in dermal fibroblasts [62]. A previous study showed there are fewer myofibroblasts in bleomycin-induced scleroderma lesions for *Smad*3-/- mice, further verifying the necessity of SMAD3 for fibroblast proliferation [63]. These results reveal a proliferation-promoting role for SMAD3 in fibroblasts. Non-canonical TGF-β signaling also participates in the proliferation of fibroblasts. For instance, TGF-β signaling activates PI3K-AKT-p21-activated kinase-2 kinase signaling in AKR-2B murine fibroblasts to increase fibroblast proliferation (Figure 1) [64]. Together, these experiments demonstrated that both TGF-β canonical and non-canonical signaling contribute to fibroblast proliferation in fibrotic diseases.

Myofibroblasts in wound healing undergo apoptosis or reverse back to the quiescent form after tissue recovery, however, FAFs are apoptosis-resistant and cannot revert to the quiescent form [36]. Tissues from human IPF show little apoptosis, but have activated phosphorylated MAP Kinases in fibroblasts [65,66]. Kulasekaran et al. revealed that p38 MAPK is necessary for TGF-β1 activation of PI3K/AKT (Figure 1), which protects normal human lung fibroblasts from apoptosis induced by Fas-caspase cascade [67]. These studies provide evidence that the apoptosis-resistant property of FAFs is related to the TGF-β non-canonical pathway in fibrotic diseases.

Overall, the proliferation/apoptosis-resistant properties of fibroblasts and the formation of myofibroblasts initiate ECM protein production in the fibrotic cascade [68]. These processes are upregulated and maintained by TGF-β signaling.

### 2.3. Paracrine TGF-β Signaling-Mediated Fibrosis

In addition to TGF-β signaling in activated fibroblasts during fibrosis, TGF-β activation in human fibrotic disease involves TGF-β signaling in non-fibroblasts, such as epithelial cells and macrophages (Figure 2). Our previous study using mice expressing human TGFβ1 under control of the keratin 5 promoter (K5.*TGF-β1* transgenic mice) to drive epidermal expression of TGFβ1, and these mice showed increased epidermal TGF-β1 levels and elevated TGFβ1 secreted by K5.*TGFβ1* keratinocytes [16]. These mice have severe skin inflammation and fibrosis characterized by myofibroblast infiltration and increased collagen accumulation [16,49], as well as increased secretion of CTGF, interleukin (IL)-1β, IL-6, interferon (IFN)-γ, and tumor necrosis factor (TNF)-α (Figure 2) [16]. Overexpression of these cytokines was reported to promote fibrosis [69,70,71,72]. Further, TGF-β1 mediated fibrosis in this mouse model was SMAD3-dependent, and topical treatment with a SMAD3 inhibitor markedly decreased fibrosis in K5.TGF-β1 transgenic mice [73]. These studies demonstrate that keratinocyte-derived TGF-β can mediate fibrosis and diverse cytokine secretions in SMAD3 dependent mechanisms.

Different SMAD proteins may exert opposing functions in epithelial cell contributions to fibrosis. In renal fibrosis, for example, activated SMAD3 promotes PAI-1 expression in human renal tubular epithelial cells, confirming SMAD3 as a pro-fibrotic mediator in epithelial cells [14]. While *Smad2* deletion in mouse tubular epithelial cells accelerated renal fibrosis by enhancing SMAD3 activation, conditional deletion of *Smad4* in mouse tubular epithelial cells significantly reduced fibrosis without affecting SMAD3 activation [74]. Overexpression of SMAD7 in rat kidney tubular epithelial cells attenuated SMAD2 activation to prevent collagen synthesis and myofibroblast generation [75]. In our K5.*TGFβ1*/K5.*Smad7* double transgenic mice, epithelial *Smad7* transgene expression reversed K5.TGF-β1 transgene-induced inflammation, fibrosis, and molecularly reduced SMAD2 and NF-κB activation [49], demonstrating that SMAD7 may abrogate fibrosis by inactivation of canonical and non-canonical TGF-β signaling.

Among immune cells, macrophage infiltration in areas of chronic inflammation is a crucial regulator of fibrosis [2,76], and macrophages are often adjacent to myofibroblasts in human pulmonary fibrosis [77], where both cell types perpetuate fibrosis. In the early inflammatory phase, macrophages in most tissues exhibit a predominantly pro-inflammatory or M1 phenotype. In the later phase of wound healing, M2-like macrophages become dominant and produce excessive pro-fibrotic factors, especially TGF-β, that foster tissue fibrosis [50]. TGF-β released by macrophages propagates myofibroblast activation [78]. Another study demonstrated that selective deletion of TGF-βR II receptor in macrophages attenuated tubulointerstitial fibrosis via a SMAD3 dependent mechanism, paralleling marked decreases of myofibroblasts and cytokines that mediate fibrosis like CTGF [79], indicating that macrophages promoted fibrosis via TGF-β activation.

In summary, TGF-β signaling is the central player of fibrotic disease as it induces myofibroblast or FAF generation, proliferation, and ECM production, as well as pro-fibrotic factors secreted by non-fibroblasts. Canonical and non-canonical TGF-β components are inter-regulated, and both participate in and exert complex effects on these cell responses. While TGF-β plays a critical role in fibrosis, it also contributes to cancer progression via CAFs, as discussed below.

## 3. Tumor Promoting Effect of TGF-β Signaling in CAFs

Fibroblasts associated with cancer are known as CAFs [36] and represent the most abundant cell type in the cancer stroma [80,81]. They are major players in cancer progression and anti-tumor therapy resistance [82,83,84,85]. CAFs harbor much higher proliferation than normal fibroblasts (NFs) [86]. Furthermore, our and many others’ studies demonstrated that CAFs promote cancer progression, whereas NFs have no or weaker effects on tumor development compared to CAFs [83,87,88,89]. TGF-β activation in the stroma indicates a poor prognosis for cancer patients [90]. In the following section, we review how TGF-β signaling influences tumor progression by regulating CAF generation and biological characteristics, and its effect on other cell types, including cancer cells and cancer stem cells in the tumor (Figure 3).

### 3.1. TGF-β Signaling and Origin of CAFs

While CAFs may be derived from several sources, one common source of CAFs is normal resident fibroblasts [91]. Resident normal fibroblasts can be induced to become CAFs via elevated TGF-β1 (Figure 3) in various cancers, including breast cancer, bladder cancer, colorectal cancer, and pancreatic cancer [92,93,94]. Elevated p-SMAD2 and p-SMAD3 were found during TGF-β1-induced CAF generation, indicating that canonical TGF-β signaling is active in this process [93]. Further, TGF-β1 alters the epigenetic signature of stromal fibroblasts leading to differential gene expressions, such as α-SMA, FAP, and stronger collagen synthesis in CAFs [95]. How every component of canonical TGF-β signaling, for example, SMAD2 and SMAD3, affects CAF formation will require additional research.

Mesenchymal stem cells (MSCs) are derived from the bone marrow, but can be recruited to tissues throughout the body and serve as another source of CAFs [96,97,98]. Quante et al. reported as many as 20% of CAFs originated from MSCs in a gastric carcinogenesis mouse model [99], revealing MSCs as an important source of CAFs. Human MSCs treated with TGF-β1 can be recruited to the tumor site and induced to express markers of CAFs in vitro by upregulation of JAK/STAT3 signaling [96,100]. Other studies have shown that TGF-β/SMAD signaling inhibition can reduce MSC-CAF transformation and abolish the protumor effects of MSCs [96,101], highlighting the critical role of TGF-β signaling on MSC-CAF mediated cancer progression.

### 3.2. The Effect of TGF-β Signaling on the Biological Properties of CAFs

CAFs are much more proliferative than their normal counterpart. Local CAF proliferation can be stimulated by TGF-β present in the TME (Figure 3). For instance, CTGF is upregulated by TGF-β and can facilitate fibroblast proliferation during fibrosis. Hepatocellular carcinoma HCC cells produce high levels of CTGF as a consequence of elevated TGF-β1 expression. Treatment with a TGF-β receptor inhibitor, LY2109761, decreased secreted CTGF; and reduced hepatocellular carcinoma growth and dissemination by inhibiting CAF proliferation [102].

In addition to regulating growth factors and CAF proliferation, TGF-β plays a critical role in inducing CAF migration (Figure 3) and contractile ability. CAFs migrate through the ECM, while dragging tumor cells via direct CAF-cancer cell contact by cell adhesion junctions [103], demonstrating that CAF migration promotes cooperative tumor cell migration and invasion. Tissue contraction mediated by CAFs is considered to be the most important cause of increased interstitial pressure, delaying drug delivery to cancer tissues [104]. Colon cancer CAF migration is enhanced by TGF-β1 in a dose-dependent manner through overexpression of tight junction protein occludin, disruption of which alleviates the migration and contractile ability of TGF-β1-stimulated CAFs [105]. Furthermore, studies showed that TGF-β secreted by cancer cells leads to activation of non-canonical TGF-β RhoA-ROCK signaling, as well as the TGF-β canonical pathway that induces transcriptional regulation of Snail1 and Twist1 target genes, finally resulting in increased contractility of CAFs [106]. Furthermore, activation of both canonical and non-canonical TGF-β pathways leads to a CAF contraction-mediated altered ECM environment, enhancing cancer cell migration and invasion [106]. These studies show that TGF-β-mediated changes in CAFs serve as a cancer cell migration/invasion signal contributing to cancer metastasis.

### 3.3. TGF-β Signaling and Metabolic Reprogramming of CAFs

Metabolic reprogramming, which fuels cell proliferation by alternative mechanisms of energy production, is a hallmark of cancer [107]. Metabolic reprogramming in CAFs is referred to as a “reversed Warburg effect (RWE),” defined by the production of more lactate, ketone bodies, and pyruvate from aerobic glycolysis [108,109] (Figure 3). In previous work, we successfully cultured human oral CAFs [110], and verified the RWE, showing that there is more lactate in the culture media of human oral CAFs, and their intracellular lactate dehydrogenase activity is higher when compared with NFs (Zhou, H. M.; et al. Reversed Warburg effect in oral cancer-associated fibroblast, unpublished; manuscript in preparation). Cancer cells can directly absorb these energy-rich metabolites, including lactate, and apply them into anabolism and proliferation [109] (Figure 3). This allows cancer cells to survive without blood vessels, as they can induce oxidative mitochondrial metabolism by absorbing metabolites produced by CAFs, illustrating how cancer cells might survive during metastasis.

TGF-β signaling regulates RWE through metabolic reprogramming related proteins. Caveolin-1 (CAV-1) is a protein predictive in human breast cancer prognosis; downregulation of CAV-1 induces metabolic reprogramming of CAFs and indicates poor prognosis in breast cancer [111,112]. TGF-β overexpression in human fibroblasts shows the phenotype of CAFs with RWE characterized by decreased mitochondrial activity and increased glycolysis via CAV-1 downregulation in the CAFs. These CAF-secreted metabolites can spread among neighboring fibroblasts and sustain the growth of breast cancer cells [112]. Another protein connecting TGF-β signaling and RWE is isocitrate dehydrogenase 3α (IDH3α). CAFs from human colon cancer samples exhibit low levels of IDH3α, which is responsible for the tumor-promoting effects of CAFs [113]. Zhang et al. found that TGF-β1-induced CAFs switch from oxidative phosphorylation to aerobic glycolysis by downregulation of IDH3α. IDH3α reduction results in accumulation of HIF-1α, promoting glycolysis of CAFs by increasing glucose uptake and upregulation of glycolytic enzymes under normoxic conditions [113], indicating that IDH3α exerts its pro-tumor role by enhancing RWE in CAFs; this process is regulated by TGF-β signaling. Future studies are needed to determine which TGF-β signaling components are involved in RWE regulation.

### 3.4. Effect of CAF-Mediated TGF-β Signaling on Cancer Progression and Therapy Resistance

TGF-β signaling is a prominent pathway in all stages of CAF-mediated cancer progression, including cancer cell proliferation, invasion, and metastasis (Figure.3). In addition to metabolic reprogramming providing an energy source to foster cancer cells, TGF-β activated CAFs secrete growth factors, such as TGF-β, FGF2, FGF7, VEGF, PDGF, and HGF, to promote cancer cell proliferation [114]. CAFs stimulated gastric cancer cell migration and invasion, which could be attenuated by *Smad2* siRNA treatment and anti-TGF-β neutralizing antibody, indicating that canonical TGF-β signaling is activated during cancer cell migration and invasion [115]. Additionally, upon activation of TGF-β signaling, CAFs synthesize more ECM protein in tumors [112]. Enhanced accumulation and re-organization of ECM can elicit a mechanically stiff microenvironment, creating a biochemical and mechanical stimulus sensed by cancer cells and/or stromal cells, leading to elevated invasion of cancer cells [116,117,118]. Moreover, secretion of IL-11 by TGF-β-stimulated CAFs triggers GP130/STAT3 signaling in colorectal cancer cells, increasing the efficiency of metastasis formation [119]. Mice treated with a TβRI-specific inhibitor LY2157299 are resistant to metastasis formation [119]. Taken together, these experiments demonstrate that TGF-β activation contributes to the tumor-promoting effect of CAFs.

TGF-β secreted by CAFs may also contribute to anti-tumor therapy resistance by influencing cellular components of a tumor, including cancer stem cells (CSCs) (Figure 3). CSCs are a minor subpopulation of cancer cells with a high-rate of self-renewal, multi-potency, and tumorigenicity [120]. CSCs have intrinsic resistance to cancer therapies through their quiescence, capacity for DNA repair, and drug transporter expression to protect them from cytotoxic therapies [121,122,123,124]. Chemotherapy can better kill proliferative cells than quiescent cells. In squamous cell carcinoma (SCC), phosphorylated SMAD2 and SMAD3 mediate the quiescent state of CSCs.TGF-β inhibition increases CSC susceptibility to chemotherapy [125]. Higher self-renewal ability (stemness) of CSCs, which means more CSCs in the tumor, indicates a higher possibility of anti-tumor therapy resistance. CAF-secreted TGF-β2 activated GLI2 (hedgehog transcription factor) in CSCs, contributing to increased stemness and intrinsic resistance to chemotherapy of CSCs [126]. Taken together, these observations suggest that both canonical and non-canonical TGF-β signaling results in anti-tumor therapy resistance by regulating CSC properties.

## 4. TGF-β Regulated Fibrotic Diseases and CAF-Mediated Cancer Progression

As discussed above, TGF-β contributes to both fibrosis and pro-tumor effects of CAFs by canonical and non-canonical signaling. FAFs in fibrosis and CAFs in cancer are induced by TGF-β and increased ECM synthesis that either leads to fibrotic diseases or promotes cancer progression. Additionally, FAFs and CAFs are both promoted by CTGF via the increased proliferation of fibroblasts. However, the role of TGF-β signaling is different in fibrosis and CAF-mediated cancer progression. First, TGF-β activation can induce CAF metabolic reprogramming that feeds metabolites to neighboring cancer cells; this is not apparent in FAFs. Second, TGF-β serves as a “bridge” between epithelial cells (keratinocytes or cancer cells) and stromal cells (FAFs or CAFs) in fibrosis and cancer. However, the crosstalk is mediated by diverse factors, resulting in a variety of consequences. For example, TGF-β could stimulate more keratinocyte growth factor secretion in myofibroblasts, and keratinocyte growth factor-stimulated keratinocytes undergo hyperproliferation [127]. But, in cancer, TGF-β/SMAD signaling promotes the secretion of IL-11 by CAFs, which triggers GP130/STAT3 signaling in tumor cells, increasing the efficiency of tumor metastasis formation [119]. These similarities and differences remind us that TGF-β signaling-mediated fibrosis and cancer progression could have an inter-regulated relationship.

Fibrosis, a disease that results in loss of organ function and sustained fibrotic responses by TGF-β activation, has been suggested to increase the risk of developing cancer [128,129]. TGF-β contributes to CAF formation, and CAFs remodel collagen to induce desmoplasia formation in the TME that is characterized by stiff and oriented collagen fibers along which tumor cells can migrate [130,131]. Matrix stiffness can further promote the proliferation of cancer cells through the Akt pathway [132]. Additionally, stiffened matrix augments fibroblast proliferation, differentiation, and activates TGF-β signaling [133]. Thus, TGF-β activation, together with CAF-fibrosis, form a positive regulation loop linking fibrosis and CAF-mediated cancer progression.

## 5. Therapeutic Targeting of TGF-β Signaling in Fibrosis and CAF-Mediated Cancer 

TGF-β signaling, a critical player in fibrosis and cancer, is increasingly being considered as a therapeutic target, due to its pro-fibrotic and tumor-promoting roles [21,134,135,136]. A wide variety of therapeutic agents that target TGF-β signaling are being developed and studied, including neutralizing antibodies that inhibit ligand-receptor binding, receptor domain-immunoglobulin fusions that trap ligands and prevent their binding to receptors, and small-molecule receptor kinase inhibitors [137].

### 5.1. TGF-β Inhibition and Fibrotic Diseases

Numerous preclinical studies showed that TGF-β inhibition exerts effective anti-fibrotic effects in various animal models across different organs [45]. One method to inhibit TGF-β signaling is to reduce TGF-β ligand generation or activation. 5-Hydroxytryptamine receptor 2B (5-HTR2B) induces transcription of TGF-β1 and activates TGF-β canonical signaling in fibroblasts. Terguride is an orally available 5-HTR2 receptor inhibitor that has anti-fibrotic effects in the SSc animal model [138]. A Phase II clinical trial showed that patients treated with terguride have decreased dermal thickness, reduced myofibroblast numbers, and lower mRNA levels of TGF-β1 and TGF-β target genes when compared with patients in the control group [139] (Table 1). TGF-β is secreted in its latent form, and in fibrotic diseases, activation of latent TGF-β occurs. Members of the integrin family play a key role in the activation of latent TGF-β. Inhibition of integrins, such as αvβ6, by neutralized antibodies, has attenuated fibrosis in animal models of kidney and lung fibrosis with less accumulation of activated fibroblasts and reduced deposition of interstitial collagen matrix [140,141]. BG00011, a humanized αvβ6-specific monoclonal antibody, has completed the Phase 2A study in IPF and demonstrated a decrease of active TGF-β signaling as evidenced by dose-dependent reductions in p-SMAD2 in bronchoalveolar lavage cells from patients without serious adverse events (NCT01371305) (Table 1), highlighting the therapeutic potential of targeting integrins and their activation of TGF-β signaling in fibrotic diseases.

A second method to inhibit TGF-β signaling is to directly target TGF-β isoforms and TGF-β receptors with monoclonal antibodies or kinase inhibitors. Fresolimumab or GC1008, a monoclonal antibody binds to and inhibits the activity of all three isoforms of TGF-β, completed a clinical trial for the treatment of resistant primary focal segmental glomerulosclerosis, a fibrotic disease, and reported one case of complete remission and two cases of partial remission of proteinuria from a total number of 16 patients [142] (Table 1). An open-label Phase I clinical trial using fresolimumab for diffuse cutaneous SSc patients yielded promising results; skin fibrosis decreased with a reduced dermal myofibroblast infiltration and inhibition of TGF-β-regulated gene expression, including CTGF [143] (Table 1). IN-1130, a selective inhibitor of TGF-βRI kinase, and GW788388, an inhibitor of both TGF-βRI and TGF-βRII kinases, both relieve renal fibrosis in the preclinical murine model [144,145] (Table 1). However, targeting TGF-β signaling at the level of TGF-β isoforms or its receptors has raised several safety concerns. For example, a percentage of fresolimumab treated patients have developed keratoacanthomas, and this adverse effect can be explained by loss of the inhibitory effects of TGF-β on keratinocyte proliferation [146]. Additionally, TGF-β inhibition (with neutralizing antibodies or other strategies) can have major cardiovascular toxicity as suggested by studies in preclinical murine models of coronary arteritis [147], aortic aneurysm [148,149], and atherosclerosis [150,151]. For instance, TGF-β blockade diminished collagen levels and destabilized the atherosclerotic plaques making them unstable or rupture-prone with higher inflammation [150,151]. Therefore, cardiovascular toxicity is the major concern of inhibition of all TGF-β isoforms or direct targeting of the TGF-β receptors. Developing more selective agents or reducing therapeutic doses of TGF-β inhibitors should be tested.

Taken together, targeting TGF-β signaling is promising for the treatment of fibrotic diseases. While not discussed here, several non-canonical TGF-β signaling interventions have entered clinical trials, and readers are pointed to a relevant review [8].

### 5.2. TGF-β Targeted Therapy and Anti-CAF-Mediated Cancer Progression 

Based on evidence showing that TGFβ can be a good therapeutic target in certain tumors, several anti-TGFβ drugs have been investigated in cancer clinical trials [152,153,154]. TGF-β activation in CAFs occurs in many cancers [90,155,156,157], however, TGF-β inhibition affects all cells in the patient, including CAFs. We will briefly discuss the effect of TGF-β inhibition on CAFs using preclinical animal or in vitro cell culture models.

LY2109761, a kinase inhibitor that blocks TβRI and TβRII kinase, can inhibit tumor growth and progression of hepatocellular carcinoma in vivo by blocking TGF-β dependent production of CTGF and CAF proliferation [102] (Table 1). LY2157299 or Galunisertib, another TGF-βRI kinase inhibitor, suppressed ovarian tumor growth partially via CAF inactivation [158] (Table 1). More recently, Yao et al. found artemisinin derivatives, a chemical isolated from Sweet Wormwood [159,160], decreased breast cancer growth, and metastasis via inactivation of CAFs through TGF-β signaling inhibition [161] (Table 1). Overall, TGF-β targeted therapy could result in reduced CAF numbers or CAF activation, thus attenuating their force as an “accomplice” in cancer, leading to an anti-tumor effect.

Clinical studies with anti-TGF-β agents have demonstrated limited, but promising, successful anti-tumor effects when they are used in combination with other cancer therapeutics. Although anti-programmed cell death protein 1(PD-1)/programmed death-ligand 1 (PD-L1) immunotherapy has achieved amazing clinical outcomes for some patients with advanced cancer, therapy resistance has raised several therapeutic challenges, including identifying additional pathways to target. TGF-β activation in the TME, especially in CAFs, has been considered a determinant of tumor T cell exclusion and poor response to PD-1/PD-L1 blockade [162,163,164]. Therefore, TGF-β inhibition in CAFs could act as a compensatory treatment with PD-1 antibody to reverse therapy resistance. This was verified in mice with progressive liver metastatic disease; the PD-1 antibody provoked a limited response, but in combination with TGF-β inhibition, a robust and enduring cytotoxic T-cell response against tumor cells occurred, preventing metastasis [162]. M7824 or Bintrafusp alfa, designed to simultaneously block the PD-L1 and TGF-β pathways, was found to be superior to TGF-β inhibition and PD-1 antibody alone to suppress tumor growth and metastasis in mouse models [165,166]. M7824 treatment not only increased the number of CD8+ T cells and NK cells, but also reduced CAF marker α-SMA expression in mouse models of breast cancer [166] (Table 1). Considering the role of CAFs in promoting metastasis, M7824 treatment may reduce metastasis, in part, through inhibitory effects on CAFs. Therefore, TGF-β inhibition can be an effective synergetic treatment with anti-PD-1/PD-L1 immunotherapy partially by targeting TGF-β-activated CAFs.

Fibrosis and CAF-mediated cancer progression are both promoted by TGF-β activation, therefore, TGF-β inhibition may have twofold benefits. This can be seen in pancreatic ductal adenocarcinoma (PDAC), where relaxin-2, an endogenous hormone, significantly inhibited TGF-β induced fibroblast differentiation into CAFs by inhibition of p-SMAD2 phosphorylation [167] (Table 1). Treatment with relaxin-2 in primary human PDAC CAFs retarded tumor growth and improved efficacy of chemotherapy in a PDAC tumor model by impairing fibrosis [167]. This study suggests that TGF-β inhibition can impede cancer progression by reducing CAF-mediated fibrosis, further increasing chemotherapeutic response by removing physical fibrotic barriers.

**Table 1 biomolecules-10-01666-t001:** Therapeutic agents that target TGF-β signaling in fibrosis and CAF-mediated cancer progression.

Agent	Target	Status	Treatment Application	Efficacy
Terguride	TGF-β1 generation and activation	Phase II clinical trial	Systemic sclerosis (SSc)	Decreased dermal thickness with reduced myofibroblast numbers [139]
BG00011	TGF-β activation	Phase II clinical trial	Idiopathic pulmonary fibrosis	Dose-dependent reductions in p-SMAD2 in bronchoalveolar lavage cells (NCT01371305)
Fresolimumab or GC1008)	TGF-β1, TGF-β2, and TGF-β3	Phase I clinical trial	Resistant primary focal segmental glomerulosclerosis	Among 16 patients, one case of complete remission and two cases of partial remission of proteinuria [142]
Fresolimumab (GC1008)	TGF-β1, TGF-β2, and TGF-β3	Phase I Clinical trial	SSc	Decreased skin fibrosis with reduced myofibroblast marker(α-SMA) expression [143]
IN-1130	TGF-βRI	Preclinical rat model study	Renal fibrosis	Decreased renal fibrosis with reduced myofibroblast marker(α-SMA) expression [144]
GW788388	TGF-βRI and TGF-βRII	Preclinical mouse model study	Renal fibrosis	Decreased renal fibrosis with reduced albuminuria [145]
LY2109761	TGF-βRI and TGF-βRII	Preclinical chick embryo model study	Hepatocellular carcinoma	Suppressed tumor growth and progression by blocking TGF-β dependent production of CTGF and CAF proliferation [102]
LY2157299 or Galunisertib	TGF-βRI	Preclinical mouse model study	Ovarian cancer	Suppressed ovarian tumor growth partially via CAF inactivation [158]
Artemisinin derivatives	p-SMAD3	Preclinical mouse model study	Breast cancer	Inhibited cancer growth and metastasis via CAFs inactivation [161]
M7824 or Bintrafusp alfa	TGF-β1, TGF-β2, TGF-β3, PD-L1	Preclinical mouse model study	Breast and colorectal cancer	Suppressed tumor growth and metastasis partially by CAF inactivation [166]
Relaxin-2	p-SMAD2	Preclinical mouse model study	Pancreatic ductal adenocarcinoma	Retarded tumor growth and improved chemotherapy efficacy by impairing fibrosis [167]

## 6. Conclusions and Perspectives

Dysregulated TGF-β activation occurs in fibrotic diseases and cancer, promoting their progression, creating a strong incentive for developing pharmacological TGFβ signaling inhibitors. Upstream and downstream TGF-β components provide various selective therapeutic targets to attenuate diverse cellular fibrotic responses.

TGF-β activated stroma, including CAFs, is a promising target for cancer. Targeting CAFs via TGF-β inhibition could decrease tumor growth and/or metastases because TGF-β activation plays a contextually important role. However, due to CAF heterogeneity with diverse functions, it is essential to target specific CAF subsets to achieve clinically relevant anti-cancer effects, hence we need to better identify specific CAF targets. Additionally, targeting TGF-β activated CAFs could be an effective method to reduce anti-tumor therapy resistance, such as anti-PD1/PDL1 immunotherapy and chemotherapy. However, some researchers found that delayed TGF-β inhibitor therapy following the anti-PD-1 escape, was better than a continuous combination of anti-PD-1 and TGF-β inhibition because of dynamic stromal changes [163]. Therefore, formulating immunotherapeutic combination regimens requires a deep understanding of the impact these agents have on the TME. To be noted, long-term inhibition of the TGF-β ligand or its receptor may cause serious adverse events, such as keratoacanthomas. Future studies to optimize regimens of TGF-β inhibitors or targeting TGF-β signaling downstream components are needed to minimize adverse effects and provide potent therapeutic effects.

## Figures and Tables

**Figure 1 biomolecules-10-01666-f001:**
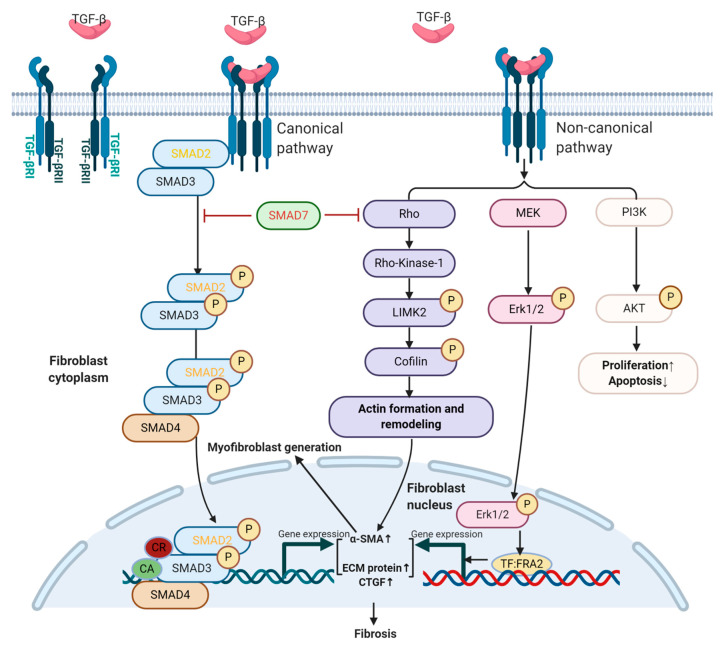
TGF-β signaling in fibroblasts. TGF-β signals start when the TGF-β ligand binds to TGF-β type II receptors (TGF-βRII), and phosphorylates TGF-β type I (TGF-βRI), activating various intracellular signaling cascades. Intracellular pathways activated by TGF-β include the canonical SMAD2/3 pathway and non-canonical TGF-β pathways, such as Rho-associated coiled-coil containing protein kinases (ROCKs), MAP kinases (ERK), and PI3K/AKT, which can regulate myofibroblast generation, fibroblast proliferation/apoptosis, and extracellular matrix (ECM) protein production. Activated small mothers against decapentaplegic homolog (SMAD) proteins or phosphorylated mediators of non-canonical signaling translocate into the nucleus and interact with DNA by transcription factors (TF), co-activators (CA, such as p53), and co-repressors (CR) to regulate gene expression, resulting in increased myofibroblast generation, elevated ECM protein production and increased pro-fibrotic factor connective tissue growth factor (CTGF) to promote fibrosis. The black characters/arrows indicate mediators/processes that promote fibrosis, and red characters indicate inhibitors of TGF-β signaling-mediated fibrosis in fibroblasts.

**Figure 2 biomolecules-10-01666-f002:**
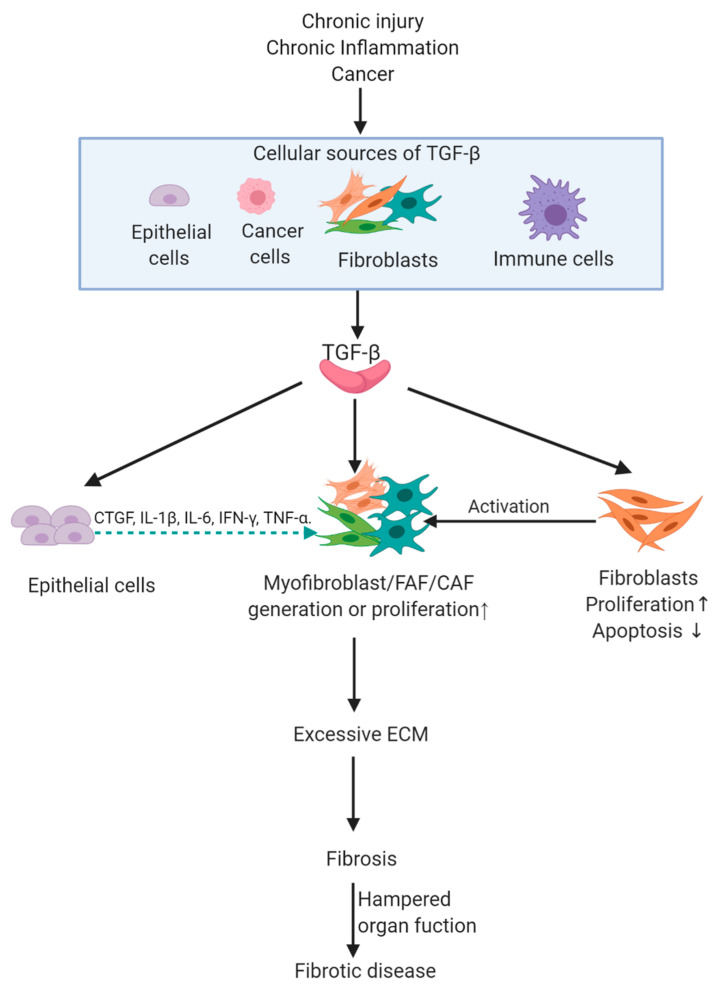
Downstream events of chronic injury/inflammation/cancer-induced TGF-β expression contributing to fibrotic disease. TGF-β is increased in the microenvironment of chronic injury/inflammation, and cancer. Once activated by TGF-β ligands, epithelial cells can secrete more fibrotic factors, such as CTGF, to increase fibroblast proliferation. Fibroblasts become activated and turn into myofibroblasts, FAFs, or CAFs, and these cells are important effector cells that can produce excessive ECM, leading to fibrosis and ultimately fibrotic diseases when organ function is hampered. The green dotted arrow indicates cytokine paracrine interaction. CTGF, connective tissue growth factor; IL-1β, interleukin-1β; IL-6, interleukin-6; IFN-γ, interferon-γ; TNF-α, tumor necrosis factor-α; FAF, fibrosis-associated fibroblast; CAF, cancer-associated fibroblast; ECM, extracellular matrix.

**Figure 3 biomolecules-10-01666-f003:**
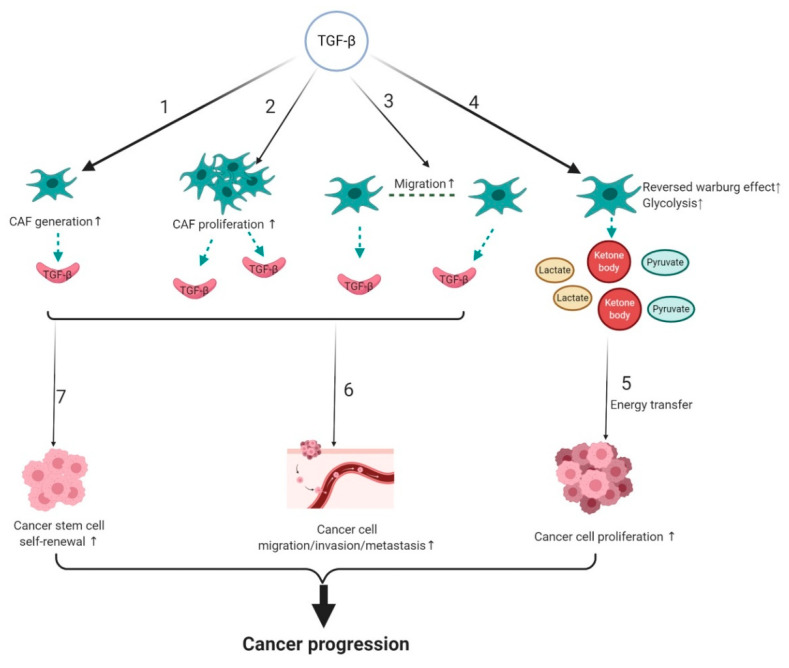
TGF-β signaling contributes to CAF-mediated cancer progression. TGF-β activation promotes 1. CAF formation; 2. CAF proliferation; 3. CAF migration; and 4. CAF glycolysis; 5. CAF energy production to fuel cancer cells. TGF-β secreted by CAFs or TGF-β activated CAFs increases 6. cancer cell migration/invasion/metastasis; and 7. cancer stem cell self-renewal ability. These processes all contribute to cancer progression. The black arrows indicate activation/promotion of the indicated cell type/process. The green dotted arrows indicate cytokine paracrine/energy production. CAF, cancer-associated fibroblast.

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
