# Peer review of "Transforming Growth Factor-β Signaling in Fibrotic Diseases and Cancer-Associated Fibroblasts"

_biomolecules, 2020, doi:10.3390/biom10121666_

Round 1
Reviewer 1 Report
This Review focuses on the crucial role that TGF-β signaling exerts in promoting fibrosis and Caf activation highlighting the TGF-β-mediated reciprocal interaction of CAFs with cancer cells.
Despite the complexity of the issue, the review is clear and covers the major points relative to the issue and indicates therapeutic opportunity targeting TGF-β signaling to treat fibrotic and cancer disease.
Below minor concerns are reported
- Figure 1 Please render more visible TGF-βRII and TGF-βRI
- In the Introduction line 73 the reference is not appropriated. You have to mention Dvorak HF. Tumors: wounds that do not heal. Similarities between tumor stroma generation and wound healing. N Engl J Med. 1986; 315:1650–1659. [PubMed: 3537791]
- Figure 2 Please check the spelling FAF/CAF ‘gegeration’ and add the main cytokines as mentioned in lines 210 and 211.
- Re the issue of cooperation between Smads and hippo signaling (line 157-160) it needs a better explanation also citing the influence of ECM stiffness and YAP/TAZ as mechanoregulators.
Reviewer 2 Report
This paper is a review article which focuses on the TGF-b signaling in fibrotic diseases and CAFs.
After a general explanation of TGF-b signaling transduction, the involvement of myofibroblast in fibrotic diseases is emphasized in the first part.
Then, the contribution of CAFs to cancer progression is mainly documented with many references, as well as a comprehensive illustration.
Finally, the development of TGF-b-targeted therapies, including information on clinical trials, is provided.
The manuscript is well organized and covers many important papers on the current topic.
Since readers will be able to update the latest findings, I recommend this manuscript for publication.
Although I carefully read through the manuscript several times, I could not find any serious flaws.
Thus, I only give minor comments.
(Fig. 1)
In the illustration, characters in “TGF-bRI” and “TGF-bRII” are too small and hardly seen.
(Fig. 2)
Smad complex is composed of 3 molecules of Smads. An illustration is miss leading.
(Fig. 2)
I could not find “myofibroblast” in the illustration.
(Line 43)
At the physiological level of expression, Samd6 inhibits BMP signaling, not TGF-b signaling.
(Line 208)
Please provide the acronym for “K5”
(2.1 Introduction to fibrotic disease)
It might be better to enumerate the name of fibrotic diseases in the first paragraph, like pulmonary fibrosis, hepatic cirrhosis. To my knowledge, glaucoma is regarded as a fibrotic disease which is associated with increased TGF-b activation. If so, this should be clearly mentioned in “2.1 introduction to fibrotic diseases”.
Reviewer 3 Report
In the manuscript "Transforming growth factor-beta signaling in fibrotic diseases and cancer-associated fibroblasts". Shi et al. provide a well thought-out and comprehensive review of the role of TGF-b signaling in fibrosis and cancer. The manuscript requires further minor revisions prior to its publication:
- The abbreviation TME (line 80) needs to be explained
- Figure 2 includes a typo ("gegeration")
- Hippo signaling should be spelled with capital letter (line 156)
- There are inconsistencies regarding the spelling of human protein names, which should be all in capital letters. For example, see lines 183-185, 217-219.
